# Associations between Cardiometabolic Risk Factors and Increased Consumption of Diverse Legumes: A South African Food and Nutrition Security Programme Case Study

**DOI:** 10.3390/nu16030354

**Published:** 2024-01-25

**Authors:** Xolile Mkhize, Wilna Oldewage-Theron, Carin Napier, Kevin Jan Duffy

**Affiliations:** 1Department of Community Extension, Mangosuthu University of Technology, Durban 4031, South Africa; 2Department of Nutritional Sciences, Texas Tech University, Lubbock, TX 79409, USA; 3Department of Sustainable Food Systems and Development, University of Free State, Bloemfontein 9301, South Africa; 4Department of Food and Nutrition, Durban University of Technology, Durban 4001, South Africa; carin.napier@auckland.ac.nz; 5School of Population Health, University of Auckland, Auckland 1023, New Zealand; 6Institute of Systems Science, Durban University of Technology, Durban 4001, South Africa; kevind@dut.ac.za

**Keywords:** cardiometabolic risk, legumes, nutritional status, elderly farmers, behavioural, diet shift, education, SDGs

## Abstract

The programme aimed to improve selected cardiometabolic risk (CMR) variables using a nutritional intervention among farmers who reported hypertensive disorders as hindrances during agricultural activities. The intervention had two case controls (*n* = 103) [experimental group-EG (*n* = 53) and control group-CG (*n* = 50)] which were tracked and whose blood pressure measurements, dietary intake, blood indices for cholesterol concentration and glucose levels from pre- and post-intervention surveys after the baseline survey (*n* = 112) were analysed. The interval for data collection was 12 weeks (±120 days) after five legume varieties were consumed between 3 and 5 times a day, and servings were not <125 g per at least three times per week. Sixty-five per cent of farmers were above 60 years old, with mean age ranges of 63.3 (SD ± 6.3) years for women and 67.2 (SD ± 6.7) for men. The post-intervention survey revealed that EG blood results indicated nutrient improvement with *p* <= 0.05 for blood glucose (*p* = 0.003) and cholesterol (*p* = 0.001) as opposed to the CG. A trend analysis revealed that cholesterol (*p* = 0.033) and systolic blood pressure (SBP); (*p* = 0.013) were statistically significant when comparing genders for all study phases. Interventions focusing on legumes can improve hypertension and cardiovascular disease and fast-track the achievement of SGDs 3 and 12 through community-based programmes.

## 1. Introduction

There has been an increasing prevalence of chronic health conditions among the elderly globally and in South Africa. The demands associated with farming activities that elderly farmers encounter often cause them to skip meals. Many often consume a diet with poor diversity, which exacerbates their already compromised health status. South Africa is a middle-income country with a high prevalence of hypertension amongst its elderly population of around 45% in 2016 [1,2,3]. Hypertension is one of the main causes of cardiovascular diseases (CVDs) [4,5]. High sodium intake, which is associated with hypertension and CVDs, is particularly prevalent in South Africa among elderly Black South African women [6,7,8]. Other hypertension-related factors include increasing age, post-menopausal conditions in women, lack of regular physical activity, a diet low in fruit and vegetables [1,9], elevated body mass index (BMI), obesity-related chronic inflammation, and metabolic co-morbidities [10].

A compromised health status associated with hypertension can exacerbate malnutrition among the elderly if not urgently prioritized [11]. Hypertension can impact their farming operations, limits their access to income, and compromises their overall well-being. Hypertension also results in various health challenges such as painful skeletal joints, cramps, fatigue, eye problems, insomnia, and memory loss, and some of these conditions are commonly associated with high morbidity and mortality rates [1,12]. The complexity of these challenges associated with hypertension can negatively impact the overall health and well-being of this population group. Therefore, if the consumption of healthy food by elderly farmers is addressed, their survival rates could be improved [7,13]. Legumes are an important source of protein; dietary fibre; and minerals such as calcium, magnesium, and iron, which are essential for human health [14]. Other legume benefits include antioxidant, antifungal, anti-microbial, anti-inflammatory, anti-hypertensive, anti-diabetic, and anti-cancer effects [15,16,17]. Regular consumption of legumes, specifically the bioactive compounds and phytochemicals present in legumes, can thus prevent age-related disorders [18,19,20] and improve cardiometabolic risk (CMR) factors [21]. Moreover, compared with animal protein sources, legumes are affordable and environmentally more sustainable to produce [15,16,17]. In addition, legumes are emerging as a sustainable food source for promoting sustainable diets in future [22,23]. 

There is an urgent need to fast-track Sustainable Development Goal (SDG)-aligned interventions to capacitate vulnerable groups, such as the elderly, to prevent hypertension-related morbidity and mortality cases in South Africa [3]. Research has revealed a strong link between health, education, and mortality in first and second world countries [24]. Therefore, the aim of this study was to develop and implement a Food and Nutrition Education Programme (FNEP) to promote the consumption of various legume varieties by elderly farmers who had chronic hypertension and high blood cholesterol and glucose levels. A key objective was to tailor a dietary intervention programme for elderly farmers using diverse legume production and consumption as a strategy to improve the health status of the participants, with specific focus on improving selected CMR variables detected in the elderly farmers in a baseline survey. In addition, the programme was aligned with the Dietary Approaches to Stop Hypertension (DASH) recommendations to determine if systolic and diastolic blood pressure would be affected by adopting a healthier dietary pattern [18,25,26]. The study envisioned that the programme would foster responsible dietary consumption and result in good health and well-being as proposed by SDGs 3 and 12. 

## 2. Materials and Methods

### 2.1. Ethical Approval

This study was supported by health care experts from the relevant local municipal clinic where the farmers obtained their chronic medications. The Research Ethics Committee of the Durban University of Technology approved the study (protocol code IREC 026/15) on 9 March 2016. The ethical protocol adhered to South African Medical Research Council and South African Human Research Council guidelines for medical research. Each participant signed a voluntary consent form after they had been briefed about the study. 

### 2.2. Study Design

This study followed a pre/post-test design, in which the intervention was informed by a cross-sectional baseline survey and was implemented in an experimental group (EG) and a control group (CG).

### 2.3. Sampling

The participant selection process was conducted at an agricultural ecology hub station (AEHS) to which the local municipality (eThekwini Municipality) allowed access to recruit all project registered members. This area was chosen because a baseline study indicated a high prevalence of cardiometabolic risk (CMR) factors such as overweight, abdominal or overall obesity, high blood pressure, high glucose and cholesterol levels, high sodium intake, and low legume consumption among elderly smallholder farmers in the area [27,28,29]. These results correlated with national data cited in the introduction [1,3].

The recruitment process was rolled out through a series of meetings with farmer organisation leadership and later with all the farmers. Inclusion criteria were farmers who gave consent and demonstrated commitment and willingness to participate in all project stages and explore the planting of legumes within their farming plots. Exclusion criteria included farmers were also part of the baseline but were not willing to give consent to commit to the intense training of the programme due to lack of time availability and uncertainty about increased legume consumption. Another exclusion criterion included farmers who were not part of the baseline survey and the municipal programme but heard about the project and arrived as walk-ins who were willing to participate during the selection process. There was strict monitoring of farmers from the CG who wanted to be moved into the EG and vice versa. The characteristics of the two sample groups (EG and CG) were similar. However, a purposive sampling strategy was employed to select the participants. The EG was selected based on the participants’ willingness to cultivate legumes and to adhere to recommended dietary guidelines required by the programme stipulations (Figure 1). The CG did not participate in the legume education programme. Addressing any legume interest among participants in the CG was reserved for the end of the intervention phase. All the recruited smallholder farmers (*n* = 103) lived in the Mariannhill area in KwaZulu–Natal, South Africa. They were part of the municipal agricultural programme and reportedly suffered from chronic hypertension. 

### 2.4. Intervention

Figure 1 above outlines the conceptual framework of the study, which was underpinned by the SDG policy framework. The study design systematically considered fundamental aspects such as the farming environment, the health and nutritional status of the elderly farmers that needed to be modified using an educational platform to positively influence behavioural shifts. The intervention study was implemented with a group of 103 participants (72 women and 31 men as gender representatives in the EG and CG). The South African Healthy Eating Guidelines were used to make recommendations for responents who were non-consumers of a balanced diet. Whilst communicating the nutritional benefits associated with legumes and dietary intake modifications for non-communicable disease management (with special focus on hypertension and diabetes). Training was also conducted on conservation agricultural practices for responsible production to increase diverse legume production, following the municipality’s agroecology guidelines aimed at addressing food and nutrition security. The data of the participants who had been involved in all the assessments were retained, and these data sets were used for all the analyses. 

Social–cognitive theory (SGT) informed the nutrition education programme content [31,32] to not only impart knowledge and skills, but also to change perceptions and fears about the affordability and cost of energy usage and the benefits of regular legume intake. Activities were formulated to support interactive educational and practical training sessions, and these were primarily aimed at increasing knowledge and facilitating social cohesion. These were important aspects to factor into the training design, as the aim was to create a community platform for farmers to share methods of practice using their hands-on training and overall training experiences. The nutrition education was conducted in the municipality’s agro-ecology hub unit and focused on the health benefits of legume consumption for managing hypertension and cholesterol levels. The training materials were developed to be user-friendly and were provided in the isiZulu language. The farmers were asked to bring notebooks for taking notes, and they were encouraged to ask questions. 

A hands-on approach was used, which implemented practical demonstrations focusing on the sensory evaluation of diverse legume dishes using standardized recipes that the farmers could easily prepare at home. The recipes assisted farmers to make it easy for them to consume legumes in a variety of ways. A variety of legume sources were introduced for the EG, which included mainly jugo beans, cow peas, red kidney beans, green mung, chickpeas, etc., which were accessible to the farmers locally for the duration of the programme for increased consumption and production. The recommended legume intake was determined using the South African Guidelines for Healthy Eating and Food Guide. The farmers had to consume 125 g, or ½ a cup, of legumes per portion 3–5 times a week over a 12-week period (3 months). This regime was similar to an earlier study that monitored consumption over the same period [33,34]. The CG (n = 50) continued their habitual dietary patterns and were not exposed to the intervention and educational programmes.

### 2.5. Data Collection

Data collection measuring instruments included a socio-demographic questionnaire which ascertained the farmers’ socio-economic status and other relevant personal information. Anthropometric measurements (height and weight) were conducted to determine body mass index (BMI). The BMI cut-off points (underweight: <18.5 kg/m^2^; normal weight: 18.5–24.9 kg/m^2^; overweight: 25–29.9 kg/m^2^; obese 1: ˃30 kg/m^2^; obese 2: ˃35 kg/m^2^; and obese 3 ˃40 kg/m^2^) [28]. Waist circumference (WC) was measured and cut-off points of ≤88 cm for women and ≤102 cm for men were used [28]. Waist-to-height ratio (WHtR) was also measured, which should be in a ratio of ≤0.5 for men and women [27,29]. Blood pressure was measured using a sphygmomanometer. All data collections were conducted once in the morning on arrival for the pre- and post-intervention phases. The South African guidelines for hypertension cut-off points were used for blood pressure analysis to determine systolic blood pressure (SBP) and diastolic blood pressure (DBP) [26]. 

Non-fasting blood was drawn using a vacutainer needle holder in a grey tube (7 mL), which was used for the blood glucose and cholesterol analyses pre- and post-intervention (after 12 weeks). A second sample of blood was drawn for a yellow tube (7 mL) for biochemical analysis. Blood samples were collected by trained phlebotomists and immediately placed in a cooler box and safely transported to the biomedical laboratory for analysis. A Biolis 15i Chemistry analyser (KAT Medical Laboratories, Roodepoort Gauteng, South Africa) was used for blood analyses. For cholesterol, ≤5.17 mmol/L reflected desirable blood cholesterol, 5.17–6.18 mmol/L reflected borderline high blood cholesterol, and ≥6.20 mmol/L reflected high blood cholesterol [35]. Glucose ranges were classified as low (<5 mmol), normal (5–6 mmol), or high (>6 mmol) [35].

Dietary intake was measured by administering a 24 h dietary intake recall questionnaire (24 h recall) as well as a food frequency questionnaire (FFQ) [36] for a 12-week period (±120 days). Data were collected pre-intervention, which was before the 12-week programme, and post-intervention (after 12 weeks) to determine legume consumption patterns. Food Finder^®^ version 3 software, based on the South African Food Composition data, was used to analyse the food consumed for nutrient intake [37]. Nutrient adequacy ratios (NARs) for macronutrients were calculated using the 24 h data and compared with dietary reference intake sources to determine the dietary intake patterns of the EG and CG [38,39]. The FFQ was used to record food group consumption consumed over one week to determine dietary diversity scores (DDSs). DDSs were calculated as an aggregate of food groups consumed over seven days (as a measure over a given time), and each food group also had to be aggregated [40,41]. Nine food groups were selected to measure DDS, as this was a relevant and a suitable standard for the nature of the study. 

The FNEP provided opportunities for reflection sessions during which the farmers could share their perceptions and experiences of including legume varieties in their diet. Feedback discussions were recorded on video, and screening these recordings facilitated discussions on some misconceptions and the lack of knowledge about legume benefits. Nuances, perspectives, and factors influencing behavioural shifts by participants were captured and considered during post-intervention sessions. 

### 2.6. Statistical Analysis

The data of the EG members who had failed to complete the full training and legume cultivation programmes and who had not arrived for blood sample collection were excluded. Data were analysed pre- and post-intervention using the IBM SPSS software program version for Windows Version 25.0. A Wilcoxon test statistic (W) was used for the same group on a single independent variable analysis, and two-way ANOVA was used to compare EG and CG pre- and post-intervention CMR variables to validate and reduce potential biases. The *p*-values for all the relevant effects (group effect—GE, time effect—TF, interaction effect—IE) were presented to determine interaction effects that might exist. Paired T-tests for statistical frequencies, variances, and correlations were also included for dietary, anthropometric and biomedical data. The Mann–Whitney U test compared differences between two independent groups, as the dependent variable, which can neither be continuous nor ordinally distributed, was used for anthropometrics data, serum cholesterol, and plasma blood glucose comparisons. A trend analysis of the EG’s CMR variables was conducted from the same EG participants who were in the baseline survey, as well as the pre- and post-intervention phases. The data were used to report on various interval results for individuals who were part of all the phases. Statistical significance was determined at *p* < 0.05.

## 3. Results

The EG mean age was 63.3 years (SD ± 6.3), compared to 67.9 years (SD ± 8.5) in the CG. The majority (65.0%) of the sample was 60 years and older, and thus categorised as elderly [42]. The study had a higher representation of women participating in the baseline, pre-intervention, and post-intervention phases, and comparisons were conducted to investigate the significance levels across the genders. Table 1 and Table 2 compare (pre- and post-) macronutrients intakes against recommended dietary allowances (RDA) for both the men and women as an observational measure from the implemented programme on dietary intake. 

The NARs revealed a higher energy intake (5939.90 ± 2313.60 kJ) by women during the post-intervention phase. This intake level contributed 92.8% of the estimated energy recommendation (EER) of 6400.80 ± 1214.64 kJ for women. Statistically significant (*p* = 0.001) higher energy intake levels were observed at the post-intervention phase for women. Although the men had a higher intake level post-intervention, it was not significant. After the post-intervention phase, women showed significant improvement for all measured macronutrient intakes of carbohydrates (*p* = 0.002), protein (*p* = 0.000), dietary fat (*p* = 0.001), and dietary fibre (*p* = 0.001), whilst no significance differences were observed for the men. Table 3 summarizes the DDSs, which did not change significantly from pre- to post-intervention for the EG or CG. However, the legume FGDS improved significantly (*p* < 0.001) from 2.4 pre-intervention to 5.7 post-intervention. The food group diversity scores (FGDSs) of three food groups were significantly lower for the EG at the post-intervention phase (for instance, dairy *p* = 0.045 and cereals *p* = 0.009) than for the CG. For the CG, eight of the nine food groups showed significantly lower FGDSs for the post-intervention phase than for the pre-intervention phase (Table 3). 

The results of the CMR variables (BMI, WC, WHtR, SBP, and DBP) for the EG are presented in Table 4 and reveal a significant per-interval (pre-intervention vs. post-intervention) comparison. No significant differences were observed between the groups, except for DBP, which was significantly lower (*p* = 0.020) for the EG during the pre-intervention phase. The Wilcoxon test initially highlighted key trends and effects of a single independent variable on the dependent variable. However, the two-way ANOVA indicated a significant difference between the EG vs. CG for SBP 0.001 (GE). The non-fasting blood results of the EG obtained post-intervention revealed that their blood glucose values had significantly improved (*p* = 0.003), whilst the EG vs. CG comparison confirmed a significant difference (*p* = 0.001 GE) for cholesterol (Table 5). 

Table 6 reveals statistically significant differences between SBP (*p* = 0.013) and cholesterol (*p* = 0.033), reporting a trend of the three study phases that included participants who were part of the baseline as well as the pre- and post-intervention surveys for the EG. The Mann–Whitney U test used for the post-intervention survey indicated no statistical significance between the genders for BMI (*p* = 0.315) and WC (*p* = 0.472). In contrast, a statistically significant (*p* = 0.004) lower mean for total cholesterol (TC) was observed for the men and women of the EG after the intervention programme (Table 5). This was confirmed by the indication that the hyperglycaemia of the EG had reduced from 38.0% to 9.0% when measured after the intervention programme. No significant difference in TC was observed for the CG. It was concluded that the consumption of legumes had significantly impacted the EG’s nutritional security and overall health when comparing the results of these variables for the two groups between the pre- and post-intervention phases.

## 4. Discussion

The study aim was to improve the health and dietary intake of the participating farmers in the EG. The study further aimed to determine the associations between cardiometabolic risk factors and increasing diverse legume consumption within a community of elderly farmers, using a synchronized thematic framework as an action track for the relevant SDGs (in Figure 1). Fundamental issues such as personal preferences, cultural sensitivity, and the need for continuous monitoring and engagement shaped the momentum for farmer participation in the intervention programme and were reflected in the SGT. A change in dietary intake was a fundamental part of the intervention programme and the legume food group as per the study objective. The study results showed significant improvements in NARs (particularly protein and fibre intake by women), FGDS (for the legume group), hypertension (DBP), cholesterol, and glucose levels within the EG, which could positively affect CMRs. This included significant differences observed between the two groups’ SBP. The two-way ANOVA analysis broadened the analytical scope to encompass the interaction between two independent variables and reaffirmed the patterns previously observed using the Wilcoxon test for an individual group analysis. Furthermore, it revealed any interaction effects that might exist and was key to comprehensively interpreting the results. The findings correlated with other studies that found that a plant-based diet incorporating legumes is rich in fibre and can improve hypertension and cardiovascular disease [43,44]. The findings of the current study thus suggest that addressing the complexity of CMR-related challenges requires extensive monitoring and that elderly farmers must be capacitated to adopt both sustainable legume production and consumption practices. Therefore, devising and reinforcing viable means of introducing legumes as a source of plant proteins that are easily accessible locally is of vital importance in transforming an elderly farming community’s health status. 

It was demonstrated that legumes are a highly viable and thus a recommended food source for the population group under study. Sharing delectable yet easy legume recipes with the farmers resulted in the acceptability of legumes in their diet, which in turn facilitated a shift in consumption patterns that influenced some CMR variables. Conducting a food and nutrition programme was a successful intervention strategy, and similar programmes may be utilised to address the many challenges and associated CMR factors among elderly smallholder farmers. A similar intervention study conducted on legume consumption revealed various benefits regarding CMRs when eating guidelines were followed and a legume intake of 400 g per week was consumed, which is a practice that should be consistently reinforced [21]. The results of the current study correlated with the FNEP, which proposes an intake of 125 g (or ½ a cup) of legumes per portion 3–5 times a day, 3 times a week. The study encouraged farmers to adopt the habit of diverse legume consumption, and this proposition was also explored by another study that demonstrated that increased legume consumption had positive outcomes over time for public health promotion as proposed by SDG 3 [45].

Earlier studies have proposed a focus on behavioural modification, as it can assist participants in addressing their nutritional challenges [31,46]. The current study achieved this objective, as it succeeded in enhancing legume consumption through the behavioural modification of farmers who had not been regular consumers of legumes. The programme prioritised the promotion of legume consumption within the existing dietary preferences of farmers by educating them on the nutritional and health benefits of increased legume intake. The project created awareness of the serious impact of hypertension and how dietary modification could improve the health status of elderly farmers who tend to spend their time in their fields and then miss clinic appointments. However, the recommendation to adopt healthier dietary choices due to the awareness of the risk of contracting noncommunicable diseases required detailed assessments of the participants’ understanding and knowledge of these risks. The literature suggests that such assessments should be supported by intensive long-term interventions that go beyond the dietary intake of legumes for health improvement, particularly when people who are burdened by overweight and obesity are involved [31,47]. 

An earlier study highlights that a dietary shift for the elderly population group is required and that education on nutrition and health-related challenges should be introduced as an intervention initiative to assist this community [48]. Because the current study’s focus on promoting legume consumption for a healthier lifestyle in the elderly correlates with the earlier South African study, which also recommends new food-based dietary guidelines (FBDGs) for the elderly to address their dietary needs, it is affirmed that the legume food group is crucial for this population group. The current study determined that foods with a high salt content were highly prevalent in the daily diets of the elderly farmers. Salt was liberally used in traditional recipes, while their cooking methods also tended to compromise the nutritional value of the food they consumed. Moreover, salt intake in their diet was not significantly reduced, as the duration of the programme was too short. It is argued that an intervention programme that facilitates the adjustment of salt intake will benefit any community. The intake of supplementary products may even be advised to further advance participants’ health and nutrition profiles in such a programme through working with local dieticians within the district. 

It was also found that limitations such as cost and legume availability hindered the regular and sustainable consumption of the legume food group. The farmers were thus encouraged not only to consume legumes but also to cultivate them to intensify crop diversification and to address food security and dietary diversification as an initiative to improve the local food system [49,50]. Other study limitations included challenges with blood pressure measurements and non-fasting blood glucose readings. This was because some participants had spent time in the field before being tested, while others had eaten before the samples were collected. BP measurements were taken before and after the dietary intervention, but many factors that can affect BP, such as age, diet, and stress, could also have impacted the findings due to the farmers’ agricultural and physical activities [45,47]. 

Another consideration associated with a combination of hypertension and obesity was to determination correlations between obesity, hypertensive disorders, and intensive therapeutic approaches [51]. Further investigation into the direct impact of hypertensive medications on blood serum is required, as this was beyond the scope of the study. This paper has shown the relevance of epidemiological studies in their role of exploring risk factors and identifying limitations particularly for vulnerable populations to implement appropriate disease control measures for CMR factors. The study results are supported by the Global Hearts Hypertension Control Program, which has been rolled out in 32 low- and middle-income countries and aimed to control risk factors such as hypertension through primary health care systems [52]. Hence, the involvement of the eThekwini local municipal clinic was vital for this study to further curb the mortality risk and addresses the study’s limitations. 

Moreover, for a more diverse investigation, future interventions of a similar nature should focus on a particular type of food while measures should be devised to limit the concurrent intake of undesirable foods containing high levels of sugar, salt, and fat. Moreover, interventions that aim to curb the uncontrollable intake of undesirable food types by elderly people with non-communicable diseases must be devised by local municipal health and agricultural systems. It is important to note that the results of the current study may not be generalised to the national population. However, the findings can be used as a reference for community-based interventions, particularly those that involve farming communities that are actively engaged in agricultural training initiatives and also grappling with hypertensive disorders. 

## 5. Conclusions

The importance of acknowledging SDGs 3 and 12 in the implemented programme provided a link between human health and environmental sustainability through the promotion of legume cultivation and consumption. The need for the wide cultivation of this food group has been affirmed as it delivers diverse health benefits in support of sustainability principles. Health and agricultural systems must engage thoroughly with this discourse to reduce the burden of disease and improve nutrition sensitive agricultural approaches by farmers who largely influence the local community. The recent impact of COVID-19 has underscored the dangers associated with cardiometabolic risk factors, as a multiplicity of the global elderly population was negatively affected by this pandemic. The results of this study support the literature by arguing that the promotion of legumes for disease prevention and nutritional benefits can help both elderly women and men in lowering their glucose and cholesterol levels. Moreover, legume consumption resulted in observable blood pressure improvement, and the study thus highlights the significant role that intervention programmes can play in improving the production and use of legumes to benefit human health. 

For future recommendations, we argue that unprecedented opportunities exist for the adoption of direct and indirect mechanisms to sustain effective health improvement projects, particularly concerning food consumption. However, existing gaps must be addressed and bridged through research to establish international and national agricultural food and health policy frameworks for implementing and monitoring effective preventive and treatment strategies. Such processes are necessary to curb cardiometabolic risk factors that ultimately lead to NCDs with disastrous consequences. 

## Figures and Tables

**Figure 1 nutrients-16-00354-f001:**
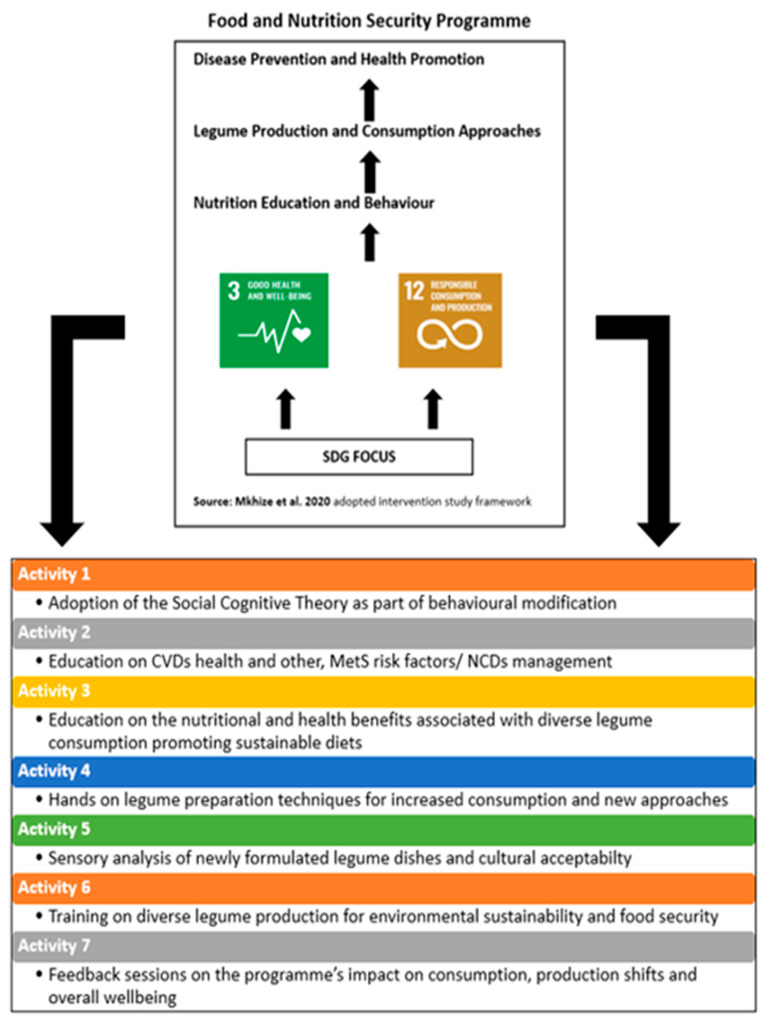
Conceptualisation of the study framework and training roll-out for each interval and activity [30] Reprinted/adapted with permission from Mkhize et al., 2020.

**Table 1 nutrients-16-00354-t001:** Pre- and post-intervention EG data for women in the EG: macronutrient adequacy ratios (NARs) and % of respondents below the DRI values (sourced from the average of three 24 h food recall surveys for women). The two groups are reported separately (EG = 39).

Nutrients/Day and DRIs for ♀ Women	Pre- EG Women Mean ± SD	Pre- EG NARs Mean % of DRI	Pre- EG % Women Consuming < 100% of DRIs	Post- EG Women Mean ± SD	Post- EG NARs Mean % of DRI	Post- EG % Women Consuming <100 of DRIs	Significance *p*-Value
Energy (kJ) EER value for active ♀ Pre-6212.05 ± 1255.75 Post- ♀6400.80 ± 1214.64	4010.83 ± 1300.89	64.6	46.2%	5939.90 ± 2313.60	97.8	41.7%	0.001
Carbohydrates (g) ♀100 EAR	139.64 ± 55.16	139.6	33.3%	177.78 ± 53.23	177.7	4.2%	0.002
Total protein (g) ♀46 RDA	32.28 ± 11.52	70.1	90.9%	59.95 ± 30.62	130.3	29.2%	0.000
Total fat (g) ♀100 EAR	24.20 ± 11.288	24.2	63.6%	43.68 ± 26.858	43.6	48.0%	0.001
Total dietary fibre (g) ♀21 AI	11.36 ± 4.56	54.0	97.4%	16.65 ± 6.46	79.2	70.8%	0.001

♀ [Women], Dietary reference intakes [DRIs]; estimated energy requirement [EER]; estimated adequacy ratio [EAR]; experimental group [EG].

**Table 2 nutrients-16-00354-t002:** Pre- and post-intervention EG data for men in the EG: macronutrient adequacy ratios (NARs) and % of respondents below the DRI values (sourced from the average of three 24 h food recall surveys for men). The two groups are reported separately (EG n = 14).

Nutrients/Day and DRIs for Men	Pre EG Men Mean ± SD	Pre EG NARs Mean % of DRI	EG % Men Consuming < 100% of DRI	Post- EG Men Mean ± SD	Post- EG NARs Mean % of DRI	EG % Men Consuming < 100% of DRI	Significance *p*-Value
Energy (kJ) EER value for active men Pre- ♂4815.55 ± 545.68 Post- ♂ 5188.34 ± 1452.19	4729.46 ± 1290.86	98.2	35.7%	5255.45 ± 1954.55	101.2	45.5%	0.408
Carbohydrates (g) ♂100 EAR	170.07 ± 53.58	170.7	0.0%	139.64 ± 55.16	139.6	33.3%	0.150
Total protein (g) ♂56 RDA	38.29 ± 16.78	68.3	92.9%	48.09 ± 20.56	85.8	72.2%	0.178
Total fat (g) ♂100 EAR	25.51 ± 12.831	25.5	85.7%	33.67 ± 16.08	33.6	50.0%	0.149
Total dietary fibre (g) ♂ 30 AI	14.44 ± 5.798	48.1	100.0%	19.93 ± 8.38	66.4	36.3%	0.054

♂ [Men].

**Table 3 nutrients-16-00354-t003:** Summary of the FGDSs for all food groups pre- and post-intervention (EG n = 53; CG n = 50).

Food Groups	Mean FGDS EG Pre Intervention	±SD	Ranges of Scores	Mean FGDS EG Post- Intervention	±SD	Ranges of Scores	Significance *p*-Value
Meat Eggs Dairy Cereals Legumes Vitamin A-rich fruit and vegetables Other fruits Other vegetables Fat and oils	6.5 1 4.3 9.6 2.4 5.0 5.3 6.6 2.1	3.07 0.00 2.18 3.67 1.35 1.80 4.02 2.28 0.90	1–4 0–1 1–9 1–5 1–7 1–8 1–19 1–16 1–5	6.1 1 3.3 7.3 5.7 3.8 5.0 6.1 1.8	2.92 0.00 2.30 4.05 2.56 1.88 2.81 3.10 0.59	1–4 0–1 1–9 1–5 1–7 1–8 1–19 1–16 1–5	0.540 1.000 0.045 0.009 0.000 0.004 0.681 0.416 0.062
FVS	42.8	18.90	23–74	40.1	20.21	23–74	0.531
DDS	8.59	±0.74		8.23	±1.11		0.097
**Food Groups**	**Mean FGDS** **CG Pre** **intervention**	**±SD**	**Range of scores**	**Mean FGDS CG Post-** **intervention**	**±SD**	**Range** **of** **scores**	
Meat Eggs Dairy Cereals Legumes Vitamin A-rich fruit and vegetables Other fruits Other vegetables Fat and oils	7.2 1 8.0 13.0 3.0 6.0 12.0 10.0 3.0	1.30 0.00 3.82 8.62 1.82 4.52 6.97 6.72 2.04	1–4 0–1 1–9 2–5 1–8 1–8 1–19 1–16 1–5	5.3 1.0 3.0 8.8 2.0 4.3 4.6 5.2 2.2	2.9 0.0 1.7 3.0 1.2 0.9 2.2 2.3 0.7	1–4 1–1 1–9 1–5 1–7 1–8 1–19 1–16 1–5	0.000 1.000 0.000 0.002 0.003 0.010 0.000 0.000 0.011
FVS	63.2	±35.81	23–75	36.4	±14.9	20–74	0.000
DDS	8.69	0.89		8.52	0.98		0.412

Dietary diversity score [DDS]; Food group diversity score [FGDS]; food variety score [FVS]; control group [CG].

**Table 4 nutrients-16-00354-t004:** Comparative data for BMI, WC, WHtR, and BP: of significance values as determined for the pre- and post-intervention surveys.

Factor Variable	EG Pre-Mean ± SD	EG Post- Mean ± SD	Significance (*p*-Values) * Pre- and Post- for EG	CG Pre- Mean ± SD	CG Post-Mean ± SD	Significance (*p*-Values) * Pre and Post for CG (*Wilcoxon Test-W*)	Significance (*p*-Values) * for the Three Relevant Effects Pre and Post for EG and CG (Two-Way ANOVA)
BMI (kg/m^2^ 18.5–24.99 (normal range)	31.07 ± 8.62	32.35 ± 6.2	0.141	31.42 ± 7.13	31.28 ± 7.32	0.298	0.385 GE; 0.399 TE; 0.951 IE
WC (cm) <102 cm♂/ >88 cm♀	99.99 ± 12.41	105.02 ± 18.59	0.218	100.98 ± 13.85	101.26 ± 14.35	0.959	0.224 GE; 0.237 TE; 0.379 IE
WHtR (cm/m^2^) (<0.5)	0.60 ± 0.10	0.65 ± 0.11	0.228	0.619 ± 0.09	0.61 ± 0.09	0.465	0.351 GE; 0.137 TE; 0.681 IE
SBP (<120 mmHg)	138.02 ± 28.41	129.58 ± 32.76	0.228	155.54 ± 28.65	155.52 ± 29.51	0.975	0.001 GE; 0.163 TE; 0.133 IE
DBP (≤80 mmHg)	84.8 ± 14.12	78.41 ± 11.53	0.020	85.50 ± 12.37	85.06 ± 12.55	0.643	0.091 GE; 0.094 TE; 0.119 IE

* The *p*-value represents the mean value when the EG and the CG are compared pre- and post-intervention body mass index [BMI]; waist circumference [WC]; waist to height ratio [WHtR]; systolic blood pressure [SBP]; diastolic blood pressure [DBP]. Group effect—GE; time effect—TE; interaction effect—IE. ♀ [Women], ♂ [Men].

**Table 5 nutrients-16-00354-t005:** A comparison between the EG and CG (pre- and post-intervention intervals to determine significance of the intervention).

Group	Pre- Intervention Mean ± SD	Post- Intervention Mean ± SD	Significance (*p*-Values) * Compared per Interval Wilcoxon Test-W)	Significance (*p*-Values) * for the Three Relevant Effects Compared per Interval (Two-Way ANOVA) for EG and CG
Glucose CG 5–6 mmol/L Glucose EG 5–6 mmol/L	6.73 ± 4.50 6.66 ± 2.84	6.94 ± 4.49 5.30 ± 2.86	0.349 0.003	0.225 GE; 0.447 TF; 0.406 IE
Cholesterol CG <5.17 mmol/L 200 mg/dL Cholesterol EG <5.17 mmol/L 200 mg/dL	6.23 ± 2.77 4.39 ± 1.42	6.30 ± 2.88 2.20 ± 0.48	0.877 0.001	0.001 GE; 0.830 TF; 0.872 IE

* The *p*-value represents the mean value when the EG and the CG are compared pre- and post-intervention. Group effect—GE; time effect—TE; interaction effect—IE.

**Table 6 nutrients-16-00354-t006:** A comparison of trend analysis data for cardio metabolic risk variables measured for the EG participants progressively at three intervals (baseline and pre- and post-intervention surveys).

CMR Variables EG Only	EG Men Mean ± SD (3 Intervals)	EG Women Mean ± SD (3 Intervals)	Significance (*p*-Values) * Gender Comparison
SBP < 120 mmhg DBP < 80 mmhg CHOLESTEROL < 5.17 mmol/L BMI kg/m^2^ (18.5–24.99) WC (cm) <102 cm♂/ > 88 cm♀	164.67 ± 4.04 84.00 ± 13.86 6.35 ± 1.11 32.40 ± 4.04 103.33 ± 9.80	134.62 ± 20.49 80.69 ± 11.50 4.51 ± 1.30 29.65 ± 6.31 98.08 ± 14.97	0.013 1.000 0.033 0.315 0.472

* The *p*-values are the mean value comparisons between women and men. ♀ [Women], ♂ [Men].

## Data Availability

Data are contained within the article.

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
