# Peer review of "Associations between Cardiometabolic Risk Factors and Increased Consumption of Diverse Legumes: A South African Food and Nutrition Security Programme Case Study"

_nutrients, 2024, doi:10.3390/nu16030354_

Round 1

Reviewer 1 Report (Previous Reviewer 1)

Comments and Suggestions for Authors

In this revision of their submission, Mkhize et al take into account the reviewers' comments and suggestions appropriately. The study is interesting with regards to its impact. The authors should empasize the epidemiological impact of such a study. Indeed, 3 of 4 people with hypertension are living in low or middle incomes contries, and among them, only one of 10 has his or her hypertension controlled. (see Moran AE, Gupta R. Implementation of Global Hearts Hypertension control programs in 32 low and middle income countries. J Am Coll Cardiol 2023; 82: 1868-84.) Accordingly a diet approach  able to lower high blood pressure (as reported by the authors) is of value.

Comments on the Quality of English Language

The study is easy to read, to understand. No drawback with regard on quality of English language

Author Response

The emphasis has been highlighted in the discussion and referenced.

Reviewer 2 Report (New Reviewer)

Comments and Suggestions for Authors

The paper describes in detail how the enforcement of legumes consumption in a special community improved CVD status. However, in my opinion, there are some issues to be solved before the paper can be accepted for publication.

Methods:

The major comments:

1/ The study outline is the comparison of two groups before and after the intervention. Thus, the appropriate statistical method to compare the effect of the intervention is two-way repeated measures ANOVA. Separate evaluation of the control and the intervention group induces a bias. Statistical evaluation, and the outcomes are to be given in results and interpreted in discussion.

2/ The study must be reproducible. Thus, sample recipes for each type of legume must be published – given in detail in supplementary file. 

Author Response

The two-way ANOVA was conducted and thank you for comment to validate the analysis, it has broadened the analytical scope to encompass the interaction between two independent variables. The findings have been reported in the results and in the discussion.

Modified recipes to English have been provided from the local IsiZulu language. Please note the idea was to promote a diverse consumption of legumes in a dish hence the recipes promote assorted legume varieties in each dish not a single legume. This was also done to only increase diversity but to also accommodate personal preferences for individuals and household members from the five legume varieties that were used in the study.

Round 2

Reviewer 2 Report (New Reviewer)

Comments and Suggestions for Authors

As suggested, the authors added some sample food preparation recipes as supplementary material.

The authors should pay more attention to how the methods and results are described. The paper will also be read by non-specialists in the field. For example, the calculation of FGDS is to be explained. In Table 3 (but not in the methods), ranges are given, but the range is lower than the mean for meat. A reader unfamiliar with the issue cannot understand it.

The authors should consult a biostatistician, as the statistical evaluation is flawed. The impact of 3 factors (groups: EG, CG; intervention: pre/post; and sex: male, female) is studied and should be evaluated using an appropriate (single) multivariate analysis. The effect of each factor and their interaction must be given. In the methods section, the statistical approach is to be clearly given (e.g., which groups were compared using the 1-way ANOVA, which fixed factors or covariates were used in 2-or 3-factor ANOVA)

If, e.g., dietary intake is given as mean±SD, then also the percentage of, e.g., pre/post out of the EER must be given as mean±SD (not a single percentage).

Tables should be understandable for the reader. For example, Tables 1 and 2 indicate 3 pre- columns and 3 post columns but only one p value. What was compared?

Table 4 is completely chaotic: how could pre- and post-values be compared using 1-way ANOVA? And then again via 2-way ANOVA without any indication of which factors were considered? Similar question for Table 5.

If trends are evaluated across 3 time intervals (Table 6), values for 3 time intervals are to be given; trends across 3 intervals cannot be presented as one mean and SD. Moreover, sex must be taken as a biological variable.

Author Response

RESPONSE TO THE REVIEWERS COMMENT

The reviewer emphasized for a two-way ANOVA analysis to be included to address potential biases hence, a two-way ANOVA was then conducted when comparing EG and CG pre and post. Table 4 and 5 has now been revised to include all three p-values for all the relevant effects (Group Effect- GE, Time Effect- TF, Interaction Effect- IE for the two way Analysis) instead of the GE which was the only value reported previously on the manuscript. The two-way ANOVA analysis for this study has simultaneously evaluated the effects of two independent variables (e.g., treatment group and time) on the dependent variables (e.g., SBP and Cholesterol etc.). It has allowed for a more nuanced understanding of these relationships, particularly in revealing any interaction effects that might exist and was key to comprehensively interpreting the results. The methods section has also clarified the distinction between these ANOVA analyses.

This manuscript is a resubmission of an earlier submission. The following is a list of the peer review reports and author responses from that submission.

Round 1

Reviewer 1 Report

Comments and Suggestions for Authors

The submission by Mkhize provides intersting results of a short intervention trial on dietetics among 120 South African farmers. They reach relevant conclusions, supported by important results : a short period of improved dietetics (12w) results in clear cut improvement in cardiometabolic risk factors among which glycemia and cholesterol levels are clearly improved. Interestingly, these improvements were more marked among women (2/3 of the included population).

The submission is well written, easy to read and to understand. The results have clinical /epidemiological implications, especially, as the authors report, for disadvantaged populations in which nutrition is usually of poor quality. This is not an African issue only, thats what is encountered in high incomes countries..with low incomes suburban populations.

The statistical analysis is appropriately designed. The reference Section is well built, with references properly selected.  

Only minor suggestions :

Abstract

Rewrite « A large majority of farmers were women with mean age ranges of 63.3 (SD ± 6.3) years and 21 67.2 (SD± 6.7) for men » , replace large majority by numbers

The Introduction and the discussion may be shortened

Author Response

Rewrite « A large majority of farmers were women with mean age ranges of 63.3 (SD ± 6.3) years and 21 67.2 (SD± 6.7) for men » , replace large majority by numbers

Addressed

The Introduction and the discussion may be shortened???

The introduction and discussion have been reviewed and also re-edited to capture all comments of the reviewers.

Reviewer 2 Report

Comments and Suggestions for Authors

The paper by MkhizeÍ“ et al. is devoted to the problem of modification of cardiometabolic risk using educational program prompting study population to consume more legumes. Theoretically, the problem is relevant, but the major concern of mine is that the article is extremely unclear. There is a lot of unnecessary information, and sometimes the point of the authors is elusive. For example, consider the sentence (lines 146-148): "During the recruitment sessions, many of the farmers indicated that they had been farming for over ten years for a living and were concerned on the impact of hypertension". What has legume production to do with cardiometabolic risk? What kind of design authors used? (the subsection in the Methods entitled "Design" is nearly empty). Overall, I would recommend that the text should be completely redone, unnessary information removed, and what is left thoroughly explained. 

Comments on the Quality of English Language

I would recommend seeking for editing assistance to make the text clearer.

Author Response

The paper by MkhizeÍ“ et al. is devoted to the problem of modification of cardiometabolic risk using educational program prompting study population to consume more legumes. Theoretically, the problem is relevant, but the major concern of mine is that the article is extremely unclear. There is a lot of unnecessary information, and sometimes the point of the authors is elusive. For example, consider the sentence (lines 146-148): "During the recruitment sessions, many of the farmers indicated that they had been farming for over ten years for a living and were concerned on the impact of hypertension". What has legume production to do with cardiometabolic risk? What kind of design authors used? (the subsection in the Methods entitled "Design" is nearly empty). Overall, I would recommend that the text should be completely redone, unnessary information removed, and what is left thoroughly explained. 

I would recommend seeking for editing assistance to make the text clearer.

Response:

  • Line 146-148 now line (164-167) Highlights the importance of understanding the need to develop an intervention project due to reported existing cardiometabolic changes faced by the farmers. The reports by farmers also highlight the motivation of shifting their existing trends of producing non-legume crops to allow their project to align towards achieving SDG 2 and SDG12. This information is farmers responsiveness to the SDGS.  
  • The study design was informed by a baseline survey conducted on the same participants.
  • A language editor has edited the manuscript.

Reviewer 3 Report

Comments and Suggestions for Authors

The aim of this study was to correct the cardiovascular risk factors identified during the baseline survey by encouraging the consumption of legumes, as well as to develop interventions using legumes as a sustainable crop that will contribute to the good health and well-being of elderly farmers suffering from hypertension, dyslipidemia and hyperglycemia.

The study is related to an important and urgent public health problem. However, I have some concerns that need to be addressed.

The introduction should include data from national studies on the prevalence of hypertension among various socio-demographic, socio-economic groups for a certain period of time. It is necessary to justify why this particular cohort – elderly farmers - was chosen.

In my opinion, the section "introduction" does not provide enough data on the benefits of eating legumes to improve cardiovascular health, hypolipidemic and hypoglycemic effects of eating legumes. The results of previous studies confirming the benefits of legumes for cardiovascular health are not presented. Have similar studies been conducted before?

Why do the authors propose to solve the problem of high incidence of arterial hypertension in the participants of the selected cohort by increasing the consumption of legumes, rather than correcting traditional risk factors and prescribing antihypertensive drugs, increasing awareness of the disease, methods of correcting risk factors?

How was the duration of exposure determined by the studied method – an increase in the consumption of legumes?

The authors did not provide a link to the results of the basic study. So the reference [27] does not correspond to the submitted text of the manuscript.

In section 2.6 – it was possible to provide a link to a questionnaire about the state of health.

In the "discussion" section, it is necessary to indicate the most important results of the study. As presented, it is too general and it is not clear what the novelty of the study is.

In addition, in my opinion, it is incorrect to analyze the results of blood tests for cholesterol and glucose, since they were not taken on an empty stomach.

Also, when interpreting the results, it is necessary to clearly indicate whether the results obtained are applicable to other socio-demographic groups of the population.

Comments on the Quality of English Language

Moderate editing of English language required

Author Response

The study is related to an important and urgent public health problem. However, I have some concerns that need to be addressed.

The introduction should include data from national studies on the prevalence of hypertension among various socio-demographic, socio-economic groups for a certain period of time. It is necessary to justify why this particular cohort – elderly farmers - was chosen.

Response:

New data has been provided.

In my opinion, the section "introduction" does not provide enough data on the benefits of eating legumes to improve cardiovascular health, hypolipidemic and hypoglycemic effects of eating legumes. The results of previous studies confirming the benefits of legumes for cardiovascular health are not presented. Have similar studies been conducted before?

Response:

There have been no specific studies on target populations that has been explained in 93-94 therefore legume interventions in South Africa are also minimal. Legume studies and benefits have been further explained 105-110.

Why do the authors propose to solve the problem of high incidence of arterial hypertension in the participants of the selected cohort by increasing the consumption of legumes, rather than correcting traditional risk factors and prescribing antihypertensive drugs, increasing awareness of the disease, methods of correcting risk factors?

Response:

The programme included building awareness of disease refer to activity 2 in Figure 1. However, considering that the participants were farmers with poor diet quality and low legume consumption trends. It was necessary to also include the legume consumption aspect while working in collaboration with the local municipal clinic to support the programme.

How was the duration of exposure determined by the studied method – an increase in the consumption of legumes?

Response:

Explanation provided in line 213-214.

The authors did not provide a link to the results of the basic study. So, the reference [27] does not correspond to the submitted text of the manuscript. In section 2.6 – it was possible to provide a link to a questionnaire about the state of health.

Response:

A link has been provided in lines 172/ 226

 In the "discussion" section, it is necessary to indicate the most important results of the study. As presented, it is too general and it is not clear what the novelty of the study is.

Response:

The discussion results study focus was more explained from line 393-396 and 402-411

In addition, in my opinion, it is incorrect to analyze the results of blood tests for cholesterol and glucose, since they were not taken on an empty stomach.

Response:

Yes, agreed it was virtually impossible to force farmers to neglect their agricultural activities in the morning and arrange for the testing. Many travel very far from the training site and some were on medication. This was the only channel that could be used to allow for participation.

Also, when interpreting the results, it is necessary to clearly indicate whether the results obtained are applicable to other socio-demographic groups of the population.

Response:

Explanation provided in lines 478-482

 Moderate editing of English language required.

Response:

A language editor has edited the document in purple.

Reviewer 4 Report

Comments and Suggestions for Authors

Review

In Abstract

“CMR variables”,  “SBP” – Please explain abbreviation when first introduced.

“The blood glucose levels of the EG were statistically significant” – what does mean- they were statistically higher/lower,  please clarify.

Please change “cholesterol” to “cholesterol concentration”.

“A trend analysis revealed that cholesterol  (p=0.033) and SBP (p=0.013) were statistically significant “ what does it mean that cholesterol was significant- please re-arrange this sentence.

“and fast-track the SGDs.” Please add explanation to SGDs.

Introduction

Line 38  to prevent hypertension-related morbidity and mortality cases [1].

In this place some numbers on prevalences should be given.

Line 41 please, exchange untimely into premature

Line 42

“Several cases of hypertensive individuals who were not aware of their status have been reported.”

Authors should address the problem  in some other way, this suggests that only some/ Few cases occurred. Probably it was not the goal.

Line 51 “intake is 5g salt [2000mg sodium” please cite that it is WHO recommendation

Lines 60-61   “An increasing prevalence in diverse community settings has indicated an increase rate of 13.5% in specific communities but has escalated to 75.5% in others, depending on 61 vulnerability levels [12].” Please, re-arrange this sentence as it is hard to read.

Line 62-3

Elderly farmers are vulnerable to hypertension and, if further ex- posed to pesticides, this can pose another threat to hypertensive people, leading to higher  blood pressure levels [13].  As this not a classic risk factor of developing hypertension and this thesis is supported in the sentence by only one reference, so it needs more references and comments at his moment if authors suggest it may be important in those communities.

Material and Methods

Line 192  “The farmers had to consume 125 g or ½ a cup of legumes 192 per portion 3-5 times a day, 3 times a week.”

It suggest to add one sentence describing what precisely types of legumes were usually consumed.  

Please, add explanations to all abbreviations when first time used throughout the text.

Also, please add explain all abbreviations under each table (including FVS, SBP, DBP and DDS  as well as EG as for a reader it is quite annoying the necessity to remember all of them at once).    

Table 4.

In cannot be named as it is now.

“Mann-Whitney U data for BMI, WC, WHtR and BP: Comparison of significance values as determined by the 344 pre-and post-intervention surveys”

Please, propose a general caption as “Comparison of anthropometric data  and blood pressure etc..? ….” 

The same for the table 5. Glucose and cholesterol levels/concentrations for…. Etc.

And for Table 6.

According to the table 5 the mean concentrations of cholesterol differed significantly before any intervention for control group and study group. Why is it so? How control group was selected? Reading the formed characteristic in Methods I thought at first they were similar. So what is the cause of that phenomena? Looking at Table 6 I guess that the reason is women generally presented lower cholesterol concentrations. Concluding there were more women in in study group than in control. It should be commented in discussion as well.

Additionally, I do not fully understand the table 6. It is entitled:

“Mann-Whitney U trend analysis data for CMR variables measured for the EG

participants progressively in three intervals (baseline and pre-and post-intervention surveys).”

It suggests baseline, pre – and post results, three measurement?

At 3 intervals? I find it not clear and in the table I just see man vs women from the study group, however I do not fully comprehend what it means. The mean from 3 measurements of blood pressure, glucose cholesterol etc or maybe the mean taken from pe and post test ?

Please clarify as to the reader this table is not clear.

Probably one of the good solution to make it all clear to the reader, and visualise baseline, intervention study and the whole process would be to create a study flow chart, with numbers of participants and men vs women including the control group.

Concisions are too long.

Please, divide it into real Conclusions and then short “Future perspectives” paragraph. I think authors include too much information in one place and real conclusions from this study are at the ending, however it is not perfect:

“The results of this  study support the literature that the promotion of legumes for disease prevention and  nutritional benefits can help both older women and men. (help in what?)

The study also highlighted the significant role that intervention programmes can play in improving the production and use of a single food such as legumes to benefit human health.   

Please, think what is really the good message for the readers. Lowering glucose level and cholesterol level basically.

Probably for the short tome of the study there was no trend toward lowering BMI and WHR, however it should be highlighted. The trend to decrees systolic blood pressure as well as significant decrease in diastolic blood pressure.

Comments on the Quality of English Language

Review

In Abstract

“CMR variables”,  “SBP” – Please explain abbreviation when first introduced.

“The blood glucose levels of the EG were statistically significant” – what does mean- they were statistically higher/lower,  please clarify.

Please change “cholesterol” to “cholesterol concentration”.

“A trend analysis revealed that cholesterol  (p=0.033) and SBP (p=0.013) were statistically significant “ what does it mean that cholesterol was significant- please re-arrange this sentence.

“and fast-track the SGDs.” Please add explanation to SGDs.

Introduction

Line 38  to prevent hypertension-related morbidity and mortality cases [1].

In this place some numbers on prevalences should be given.

Line 41 please, exchange untimely into premature

Line 42

“Several cases of hypertensive individuals who were not aware of their status have been reported.”

Authors should address the problem  in some other way, this suggests that only some/ Few cases occurred. Probably it was not the goal.

Line 51 “intake is 5g salt [2000mg sodium” please cite that it is WHO recommendation

Lines 60-61   “An increasing prevalence in diverse community settings has indicated an increase rate of 13.5% in specific communities but has escalated to 75.5% in others, depending on 61 vulnerability levels [12].” Please, re-arrange this sentence as it is hard to read.

Line 62-3

Elderly farmers are vulnerable to hypertension and, if further ex- posed to pesticides, this can pose another threat to hypertensive people, leading to higher  blood pressure levels [13].  As this not a classic risk factor of developing hypertension and this thesis is supported in the sentence by only one reference, so it needs more references and comments at his moment if authors suggest it may be important in those communities.

Material and Methods

Line 192  “The farmers had to consume 125 g or ½ a cup of legumes 192 per portion 3-5 times a day, 3 times a week.”

It suggest to add one sentence describing what precisely types of legumes were usually consumed.  

Please, add explanations to all abbreviations when first time used throughout the text.

Also, please add explain all abbreviations under each table (including FVS, SBP, DBP and DDS  as well as EG as for a reader it is quite annoying the necessity to remember all of them at once).    

Table 4.

In cannot be named as it is now.

“Mann-Whitney U data for BMI, WC, WHtR and BP: Comparison of significance values as determined by the 344 pre-and post-intervention surveys”

Please, propose a general caption as “Comparison of anthropometric data  and blood pressure etc..? ….” 

The same for the table 5. Glucose and cholesterol levels/concentrations for…. Etc.

And for Table 6.

According to the table 5 the mean concentrations of cholesterol differed significantly before any intervention for control group and study group. Why is it so? How control group was selected? Reading the formed characteristic in Methods I thought at first they were similar. So what is the cause of that phenomena? Looking at Table 6 I guess that the reason is women generally presented lower cholesterol concentrations. Concluding there were more women in in study group than in control. It should be commented in discussion as well.

Additionally, I do not fully understand the table 6. It is entitled:

“Mann-Whitney U trend analysis data for CMR variables measured for the EG

participants progressively in three intervals (baseline and pre-and post-intervention surveys).”

It suggests baseline, pre – and post results, three measurement?

At 3 intervals? I find it not clear and in the table I just see man vs women from the study group, however I do not fully comprehend what it means. The mean from 3 measurements of blood pressure, glucose cholesterol etc or maybe the mean taken from pe and post test ?

Please clarify as to the reader this table is not clear.

Probably one of the good solution to make it all clear to the reader, and visualise baseline, intervention study and the whole process would be to create a study flow chart, with numbers of participants and men vs women including the control group.

Concisions are too long.

Please, divide it into real Conclusions and then short “Future perspectives” paragraph. I think authors include too much information in one place and real conclusions from this study are at the ending, however it is not perfect:

“The results of this  study support the literature that the promotion of legumes for disease prevention and  nutritional benefits can help both older women and men. (help in what?)

The study also highlighted the significant role that intervention programmes can play in improving the production and use of a single food such as legumes to benefit human health.   

Please, think what is really the good message for the readers. Lowering glucose level and cholesterol level basically.

Probably for the short tome of the study there was no trend toward lowering BMI and WHR, however it should be highlighted. The trend to decrees systolic blood pressure as well as significant decrease in diastolic blood pressure.

Author Response

ABSTRACT

“CMR variables”,  “SBP” – Please explain abbreviation when first introduced.

Response:

 Addressed in line 16.

“The blood glucose levels of the EG were statistically significant” – what does mean- they were statistically higher/lower, please clarify.

Response:

 Addressed.

Please change “cholesterol” to “cholesterol concentration”.

Response:

 Addressed line 20.

 “A trend analysis revealed that cholesterol  (p=0.033) and SBP (p=0.013) were statistically significant “what does it mean that cholesterol was significant- please re-arrange this sentence.

Response:

 Addressed line 27.

“and fast-track the SGDs.” Please add explanation to SGDs.

Response:

  Addressed line 31.

INTRODUCTION

Line 38 to prevent hypertension-related morbidity and mortality cases [2]. In this place some numbers on prevalences should be given.

Response:

 Numbers are now added in line 41 and the now references 2 is supporting statement.

Line 41 please, exchange untimely into premature.

Response:

Added in line 53.

Line 42

“Several cases of hypertensive individuals who were not aware of their status have been reported.” Authors should address the problem in some other way, this suggests that only some/ Few cases occurred. Probably it was not the goal.

Response:

 The problem was addressed in the project and explained in line 52-53.

 Line 51 “intake is 5g salt [2000mg sodium” please cite that it is WHO recommendation.

Response:

 Refence added as reference 12.

Lines 60-61 “An increasing prevalence in diverse community settings has indicated an increase rate of 13.5% in specific communities but has escalated to 75.5% in others, depending on 61 vulnerability levels [12].” Please, re-arrange this sentence as it is hard to read.

Response:

 Addressed in line 70-71

Line 62-3

Elderly farmers are vulnerable to hypertension and, if further ex- posed to pesticides, this can pose another threat to hypertensive people, leading to higher blood pressure levels [13].  As this not a classic risk factor of developing hypertension and this thesis is supported in the sentence by only one reference, so it needs more references and comments at his moment if authors suggest it may be important in those communities.

Response:

 There is limited research conducted on the farmer. References added.

MATERIAL AND METHODS

Line 192 “The farmers had to consume 125 g or ½ a cup of legumes 192 per portion 3-5 times a day, 3 times a week.” It suggest to add one sentence describing what precisely types of legumes were usually consumed.

Response:

Sentence aligned line 213-214

Please, add explanations to all abbreviations when first time used throughout the text.

Also, please add explain all abbreviations under each table (including FVS, SBP, DBP and DDS as well as EG as for a reader it is quite annoying the necessity to remember all of them at once).    

Response:

 Explanations are provided below the tables.

Table 4.

In cannot be named as it is now.

“Mann-Whitney U data for BMI, WC, WHtR and BP: Comparison of significance values as determined by the 344 pre-and post-intervention surveys” Please, propose a general caption as “Comparison of anthropometric data  and blood pressure etc..? ….” 

Response:

The variables were identified and categorized in the same way even on the first section of the measuring instrument (health questionnaire). The table name change will be coherent with methodology.

The same for the table 5. Glucose and cholesterol levels/concentrations for…. Etc. And for Table 6.

Response:

The same response stated above applies for the rest of the tables.

According to the table 5 the mean concentrations of cholesterol differed significantly before any intervention for control group and study group. Why is it so? How control group was selected? Reading the formed characteristic in Methods I thought at first they were similar. So what is the cause of that phenomena? Looking at Table 6 I guess that the reason is women generally presented lower cholesterol concentrations. Concluding there were more women in in study group than in control. It should be commented in discussion as well.

Response:

The was no statistical difference for the CG (Pre 6.23 ± 2.77 Vs Post 6.30 ± 2.88 = 0.906). Correct they were all part of the baseline. The CG were members also from the baseline who did not want to engage in the intervention due to lack of interest and willingness to adhere to the specified guidelines. Women were in the majority for both groups.

Additionally, I do not fully understand the table 6. It is entitled:

“Mann-Whitney U trend analysis data for CMR variables measured for the EG

participants progressively in three intervals (baseline and pre-and post-intervention surveys).”It suggests baseline, pre – and post results, three measurement?

Response:

Yes, that is current it looks at all the participants who started in the baseline and were part of the EG.

At 3 intervals? I find it not clear and in the table I just see man vs women from the study group, however I do not fully comprehend what it means. The mean from 3 measurements of blood pressure, glucose cholesterol etc or maybe the mean taken from pe and post test ? Please clarify as to the reader this table is not clear.

Response:

The 3 intervals refers to the same group measure according to the response above (baseline, pre-intervention and post intervention).

Probably one of the good solution to make it all clear to the reader, and visualise baseline, intervention study and the whole process would be to create a study flow chart, with numbers of participants and men vs women including the control group.

Response:

 Clarity has been provided in line 319-320.

Concisions are too long. Please, divide it into real Conclusions and then short “Future perspectives” paragraph. I think authors include too much information in one place and real conclusions from this study are at the ending, however it is not perfect:

Response:

 The conclusions have been readjusted refer to 489-496/ 499.

“The results of this study support the literature that the promotion of legumes for disease prevention and nutritional benefits can help both older women and men. (help in what?).

Response:

 Addressed 495-496.

The study also highlighted the significant role that intervention programmes can play in improving the production and use of a single food such as legumes to benefit human health.  Please, think what is really the good message for the readers. Lowering glucose level and cholesterol level basically.

Response:

This is addressed in 494-495.

Probably for the short tome of the study there was no trend toward lowering BMI and WHR, however it should be highlighted. The trend to decrees systolic blood pressure as well as significant decrease in diastolic blood pressure.

Response:

This is addressed in 494 -496.

A language editor has edited the manuscript.

Round 2

Reviewer 2 Report

Comments and Suggestions for Authors

I have no further comments.

Author Response

The manuscript has now been further revised with the other reviewers comments addressed and highlighted in red. 

Reviewer 3 Report

Comments and Suggestions for Authors

We would like to note that the authors have done significant work to improve the article. However, some questions remain:

The introduction section provides a significant amount of data on already well-studied nutritional factors, such as table salt, but in my opinion does not provide enough data on the benefits and effects of legumes, which are emphasized. 

Also the introduction contains questionable conclusions such as lines 85-86, 89-92, 115-116, 454-456.

Also in section 2.5, the authors give a lot of general phrases about the educational process, but do not provide specific practical recommendations on nutrition, lifestyle, etc. - lines 185-192.

Author Response

The introduction section provides a significant amount of data on already well-studied nutritional factors, such as table salt, but in my opinion does not provide enough data on the benefits and effects of legumes, which are emphasized. 

This has been revised in pages 73-76.

the introduction contains questionable conclusions such as lines 85-86, 89-92, 115-116, 454-456.

Addressed please refer to page 89, 96-97, 121, 461.

Also in section 2.5, the authors give a lot of general phrases about the educational process, but do not provide specific practical recommendations on nutrition, lifestyle, etc. - lines 185-192.

Addressed in 187-193.

Reviewer 4 Report

Comments and Suggestions for Authors

The paper is much improved.

I am not sure if I read what authors meant “under the Agroecology Unit” , if it was not about Ministry of Agriculture.

Again, I did not get the information what types of legumes farmers did consume?

We are not aware which of them are available in the country where the studies were conducted.

As a cardiologist once again I highlight that pesticides use is not t a classic risk factor for hypertension and the reference used by authors is rather week -one article on children.

Please, change the statement into lighter one as it is in other medical articles such as” occupational exposure to pesticides may play a role in the development of cardiovascular diseases” including hypertension, however this should be supported by 2-3 serious reports.

In tables 5/6  CMR should be absolutely explained as in cardiology. Medicine this abbreviation means something different and it may be confusing.

And I still continue thinking that entitling the tables 4/5/6  as” Mann-Whitney U data for BMI, WC, WHtR and BP..” is a not a recommended form. Normally, we express it as comparison of….etc.

As an example I enclose the patter article below.

By the way,  authors also add appendix with abbreviations used and in my opinion it would be very helpful.

https://www.ncbi.nlm.nih.gov/pmc/articles/PMC6706911/

Comments on the Quality of English Language

The article is much improved.

However, reading it still I have the feeling that it should be peered out by the English native user experienced in scientific language.

Sometimes I am not sure if I read what authors meant for example “under the Agroecology Unit” , if it was not about Ministry of Agriculture etc.

Author Response

I am not sure if I read what authors meant “under the Agroecology Unit” , if it was not about Ministry of Agriculture.

In South Africa the national department of agriculture is more focused on rural development and the metros are more aligned to the municipality programmes for agriculture. Therefore, the study was conducted in the eThekwini Metropolitan and eThekwini Municipality has set up Agroecology zones responsible for agro-ecological practices, agribusiness and local economic development. Hence the study was conducted in one of the Agroecology zones called Marianhill Agricultural Hub and was supported by eThekwini Municipality and not the national department of agriculture.

Again, I did not get the information what types of legumes farmers did consume?

Addressed 213- 215

We are not aware which of them are available in the country where the studies were conducted.

 Addressed 213- 215

As a cardiologist once again I highlight that pesticides use is not t a classic risk factor for hypertension and the reference used by authors is rather week -one article on children.

Reference removed.

Please, change the statement into lighter one as it is in other medical articles such as” occupational exposure to pesticides may play a role in the development of cardiovascular diseases” including hypertension, however this should be supported by 2-3 serious reports.

 Reference removed.

In tables 5/6  CMR should be absolutely explained as in cardiology. Medicine this abbreviation means something different and it may be confusing.

  Titles revised

And I still continue thinking that entitling the tables 4/5/6  as” Mann-Whitney U data for BMI, WC, WHtR and BP..” is a not a recommended form. Normally, we express it as comparison of….etc.

 Titles revised

 As an example, I enclose the patter article below. By the way, authors also add appendix with abbreviations used and in my opinion it would be very helpful.

 An abbreviation list has been added pg. 547-562. Thank you for the suggestion.

However, reading it still I have the feeling that it should be peered out by the English native user experienced in scientific language.

The article has been peered reviewed by a native English editor who has experience with scientific writing a memorandum has been provided.